# MILCA: Malaria Parasite Detection from Sample-Level Weak Labels

**Petru Manescu** [ORCID]                 P.MANESCU@UCL.AC.UK
**Delmiro Fernandez-Reyes**      DELMIRO.FERNANDEZ-REYES@UCL.AC.UK
*Department of Computer Science, University College London*

**Editors:** Accepted for publication at MIDL 2026

## Abstract

Malaria diagnosis requires the inspection of multiple image fields per sample. Training vision models for malaria parasite detection typically requires large numbers of expert-provided bounding boxes, which are costly to obtain and often impractical in real-world deployments. We introduce MILCA, a weakly supervised object detection framework that learns parasite localization from sample-level diagnostic labels, which are routinely recorded in clinical practice. MILCA combines Multiple Instance Learning (MIL) for sample classification with an iterative Class Activation (CA) Mapping procedure that yields coarse parasite pseudo-labels, which are further enriched with hard negatives from parasite-free samples. These pseudo-labels enable training a detector without any manual bounding-box supervision. Experiments on multiple microscopy datasets show that MILCA achieves reliable detection and counting performance under fully weak supervision, and that fine-tuning with only a small fraction of expert annotations provides substantial additional gains, outperforming supervised and pseudo-labeling baselines under the same or lower annotation budgets. By converting coarse, sample-level clinical labels into effective object-level supervision, MILCA provides a label-efficient route toward automated malaria parasite detection and a general approach for weakly supervised blood film analysis.

**Keywords:** Malaria, Weak Supervision, Object Detection, Multiple Instance Learning, Class Activations

## 1. Introduction

Malaria remains a major global health challenge, disproportionately affecting children in low- and middle-income countries (LMICs) of the Global South (Organization et al., 2022). The current gold diagnosis standard is the visual inspection of Giemsa-stained Thick Blood Films (TBFs). Microscopists must identify extremely small, faint parasites among numerous artefacts such as platelets or staining debris, a demanding and time-consuming process that is difficult to scale in resource-limited regions (Organization and for Disease Control, 2010). Parasite density estimation further guides prognosis and treatment efficacy (Siahaan, 2018). Automated detection systems based on deep learning offer promise for high-throughput TBF analysis (Mehanian et al., 2017; Torres et al., 2018; Yang et al., 2019; Manescu et al., 2020b,a; Chibuta and Acar, 2020; Kassim et al., 2021), yet nearly all rely on tens of thousands of manually annotated bounding boxes, and on segmentation-based region proposal techniques, an impractical requirement for widespread deployment.

By contrast, *sample-level* diagnostic labels (malaria-positive vs. malaria-negative) are routinely collected as part of standard clinical workflows and are inexpensive to acquire.

TBF samples consist of collections or *bags* of image fields, naturally suited to a Multiple Instance Learning (MIL) formulation. MIL has been widely used in computational pathology to learn from slide-level labels (Campanella et al., 2019; Gadermayr and Tschuchnig, 2024) and has seen limited use in hematology (Sidhom et al., 2021; Manescu et al., 2023; Gao et al., 2023), but has not been explored for parasitic infection detection in TBFs, where object instances are tiny, numerous, and visually ambiguous.

MIL alone, however, produces only sample-level predictions and does not yield object-level localization, which is essential for parasite density estimation and clinical interpretability. Weakly supervised object detection (WSOD) methods based on Class Activation Maps (CAMs) have been used to derive localization cues from image-level labels (Zhang et al., 2021; Belharbi et al., 2025; Kniesel et al., 2025), but they have not been investigated for high-magnification bright-field TBF microscopy. Moreover, existing WSOD pipelines assume that each labeled image contains at least one target instance, whereas a malaria-positive sample may contain hundreds of fields and only a small fraction with parasites. For example, (Kniesel et al., 2025) assumes image-level labels for single electron-microscopy images, a modality not used in clinical workflows. In contrast, MILCA operates on clinically acquired brightfield thick blood films, where each sample consists of up to 100 fields and requires a MIL backbone rather than a single-image classifier to account for the sample-level diagnostic labels routinely produced in practice.

In addition to weakly supervised methods, semi-supervised learning (SemiSL) has been widely investigated to reduce annotation costs in medical imaging. SemiSL frameworks leverage both limited labeled data and large pools of unlabeled images, tipically through consistency regularization, pseudo-labeling, or teacher–student schemes (Chen et al., 2022; Bai et al., 2017; Madani et al., 2018; Xu et al., 2022; Shokrollahi et al., 2024). However, these methods require an initial supervised detector, a challenge in malaria microscopy, where only a handful of bounding boxes may be available. MILCA circumvents this limitation by constructing the initial detector entirely from weak sample-level supervision before incorporating limited strong labels. Related efforts have explored self-supervised representation learning and domain adaptation for microscopy analysis, aiming to improve robustness across imaging conditions or reduce labeled data requirements (Dilawar et al., 2025).

Our main contributions are:

1. **MIL for malaria diagnosis.** We formulate thick blood film analysis as a Multiple Instance Learning problem operating directly on bags of high-magnification fields, enabling malaria positivity prediction from sample-level labels without candidate extraction, handcrafted preprocessing, or ROI selection.

2. **ICAM: a domain-adapted iterative CAM refinement.** Building on WSOD principles but adapted to small, dense malaria parasites in TBF, ICAM combines confidence-coupled erasure, decayed CAM accumulation, and hard-negative mining to obtain usable pseudo-annotations from MIL predictions.

3. A hybrid weak+strong supervision regime with strong generalization and high label efficiency. We show that detectors initialized purely from MILCA pseudo-annotations and fine-tuned with only a small fraction of manual labels outperform supervised and simple SemiSL baselines under the same or even lower—annotation budgets. A

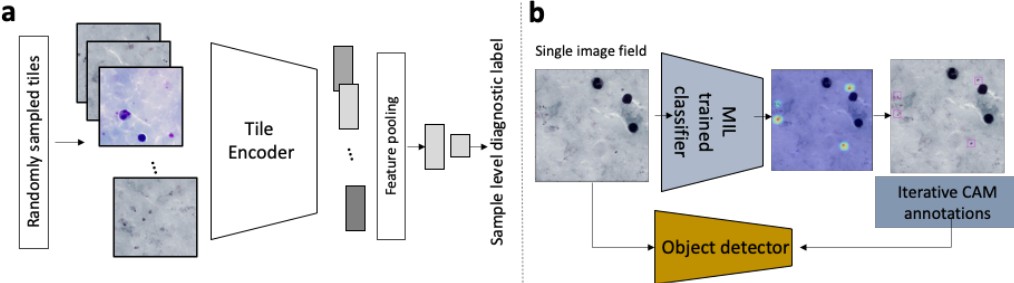

Figure 1: Brief overview of MILCA. (a) Multiple Instance Learning for TBF sample classification. (b) CAM-based generated annotations are used to train an object detector.

multi-center evaluation further demonstrates robust performance and improved label efficiency.

## 2. Methods

MILCA converts sample-level diagnostic labels into object-level supervision through three stages: (i) multiple instance learning (MIL) for sample classification, (ii) iterative CAM refinement (ICAM) for pseudo-annotation generation, and (iii) pseudo-label–driven detector training with optional fine-tuning (Fig. 1).

### 2.1. Datasets

We trained and evaluated MILCA on three thick blood film (TBF) microscopy datasets collected from different geographic and clinical settings (Table 1). The Ibadan dataset (Manescu et al., 2020b) contains 144 malaria-negative and 155 malaria-positive samples, each with approximately 100 high-resolution image fields. The Chittagong-1 dataset (Kassim et al., 2021) comprises 50 negative and 150 positive samples, with 20 image fields per sample. Finally, the Chittagong-2 dataset (Yang et al., 2019) includes 150 positive samples with on average 12 fields per sample and over 80k expert-annotated parasites.

The Ibadan dataset was acquired using a laboratory-grade digital scientific camera (PCO Edge) mounted on a conventional light microscope, while the Chittagong datasets were acquired using smartphone-based microscopy attachments, resulting in differences in image appearance and color. The Chittagong-2 dataset was acquired for a different study and contains only malaria-positive samples with higher parasite density annotations than Chittagong-1 (Yang et al., 2019) and we used it exclusively for evaluating object-level parasite detection and counting. This dataset was fully held out from all training stages, including MIL training, ICAM pseudo-annotation generation, and detector initialization. All three datasets were imaged under 100× oil-immersion objectives.

Table 1: Summary of datasets used for training and evaluation. *Used only for evaluation of the parasite detection. **Image field size containing TBF is of approximately 2400x2400; the rest is black background

| Dataset | Ibadan (Manescu et al., 2020b) | Chittagong-1 (Kassim et al., 2021) | Chittagong-2* (Yang et al., 2019) |
|---|---|---|---|
| **Negative Samples** | 144 | 50 | 0 |
| **Positive Samples** | 155 | 50 | 150 |
| **Image fields per sample** | 100 | 20 | 12 |
| **Image field size (pixels)** | 2560x2160 | 4032x3024** | 4032x3024** |
| **No. annotated parasites** | 2,986 | 25,765 | 84,961 |
| **Camera** | PCO Edge | Smartphone | Smartphone |
| **Country** | Nigeria | Bangladesh | Bangladesh |

## 2.2. Sample-level predictions with Multi-Instance Learning

Each TBF sample is treated as a *bag* $B = \{x_1, x_2, \ldots, x_n\}$ of image patches (instances) randomly cropped from digitized image fields acquired under a 100x 1.4 NA objective. According to MIL assumptions:

$$y_B = \begin{cases} 1 & \text{if at least one instance } x_i \text{ in bag } B \text{ contains malaria parasites} \\ 0 & \text{if all instances in bag } B \text{ are negative.} \end{cases}$$

Each tile $x_i$ is passed through a ConvNet encoder (Simonyan and Zisserman, 2014) $f_\theta(\cdot)$, followed by global average pooling (GAP), producing a feature embedding

$$h_i = \text{GAP}(f_\theta(x_i)), \quad h_i \in \mathbb{R}^d. \tag{1}$$

The bag-level representation is obtained via max feature pooling:

$$H(B) = \max_{i=1,\ldots,n} h_i. \tag{2}$$

A linear classifier $g_\phi(\cdot)$ then predicts the bag label probability:

$$p(y_B = 1 \mid B) = \sigma\big(g_\phi(H(B))\big), \tag{3}$$

where $\sigma(\cdot)$ is the sigmoid function. For each sample, we construct a bag of 50 tiles. When a sample contains $F$ image fields ($F = 20$ for Chittagong-1; $F = 100$ for Ibadan), we first extract one random tile from each available field. If $F < 50$, the remaining $(50 - F)$ tiles are obtained by sampling fields uniformly with replacement, allowing the same field to be selected multiple times and extracting additional random crops from the selected fields.

We adopted max-pooling to aggregate field embeddings into a bag representation since, in many cases, only a small fraction of image fields in a positive sample contain parasites. An ablation comparing max pooling and attention-based MIL aggregation (Ilse et al., 2018) showed comparable overall performance, with max pooling yielding slightly higher recall and attention pooling favouring precision; details are provided in Appendix C.

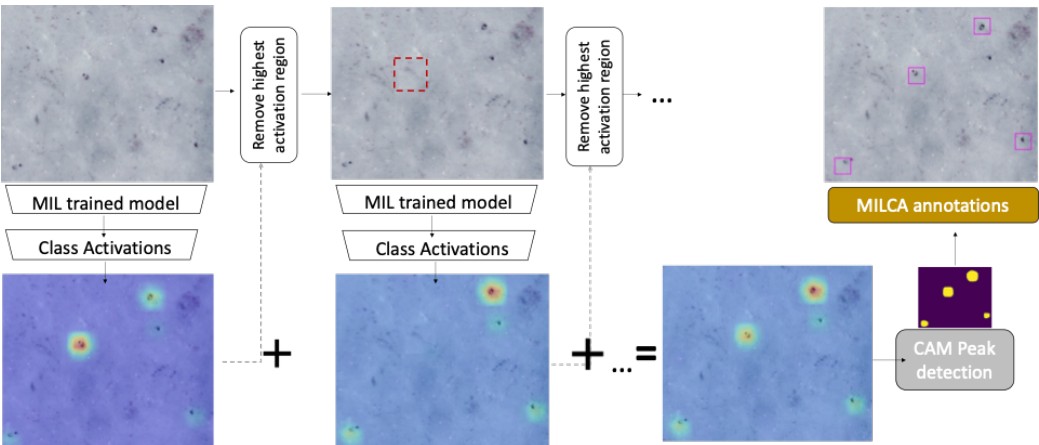

Figure 2: Iterative Class Activation Maps for Malaria Parasite Annotations Generation (ICAM).

The MIL encoder is initialized from ImageNet-pretrained VGG-19 weights and trained end-to-end without freezing. Binary cross-entropy loss was used to optimize the model weights. At test time, all available image tiles from each sample were passed through the trained model, and their aggregated bag representation $H(B)$ was used to predict the sample label. The model was fully trained for 100 epochs with an SGD optimizer and a learning rate of 0.0003. The resulting MIL model provided two key signals: (i) sample-level predictions and (ii) per-field confidence scores used to trigger ICAM refinement.

### 2.3. Pseudo-annotation generation with Iterative Class Activations

While MIL enables sample-level classification, it does not localize individual parasites. To generate pseudo-annotations, for each high-confidence malaria-positive image field ($p(y_B) >$ 0.9), we derive Class Activation Maps (CAMs) from the MIL classifier. Given convolutional feature maps $F \in \mathbb{R}^{C \times H \times W}$ and classifier weights $w_{c,k}$ for class $c$, the CAM is computed as

$$\mathrm{CAM}_c(i, j) = \sum_{k=1}^{C} w_{c,k} \, F_k(i, j). \tag{4}$$

CAMs highlight discriminative image regions that contribute to the positive class, but often emphasize only the most prominent parasite or artifact. To address this, we introduce Iterative Class Activation Mapping (ICAM), which progressively reveals multiple faint parasite instances through a sequence of erasure and refinement steps. At each iteration $s$, the highest CAM region is erased from the image $I^s$, and a new CAM is computed if the classifier score of the modified image remains above a confidence threshold (empirically, $\tau = 0.7$). At each iteration, the ICAM map is updated by adding the new CAM with an exponential decay factor ($\delta = 2$), ensuring that earlier discoveries retain more influence:

---

**Algorithm 1:** Iterative Class Activation Mapping (ICAM)

---

**Require:** Image $I$, trained MIL classifier $f_\theta$, confidence threshold $\tau$, decay factor $\delta$,
   maximum iterations $S$

**Ensure:** ICAM heatmap

  1: Initialize ICAM $\leftarrow 0$, $s \leftarrow 0$
  2: **while** $f_\theta(I) > \tau$ **and** $s < S$ **do**
  3:    Compute $\text{CAM}_s \leftarrow \text{CAM}(I)$
  4:    Update ICAM $\leftarrow$ ICAM $+ \delta^{-s} \cdot \text{CAM}_s$
  5:    $I \leftarrow$ erase_highest_activation$(I, \text{CAM}_s)$
  6:    $s \leftarrow s + 1$
  7: **end while**
  8: **return** ICAM

---

$$\text{ICAM}(I) = \text{CAM}(I) + \sum_{s=1}^{S} \delta^{-s} \cdot \text{CAM}(I^s). \tag{5}$$

To convert ICAM maps into bounding boxes, we normalized each map, apply Otsu thresholding to isolate salient regions, eroded small artifacts with a 5x5 kernel, and detected subsequent peaks separated by at least 10 pixels. Around each peak, a bounding box of fixed size (64×64 pixels) is placed. This box dimension reflects typical parasite size in TBF microscopy (Manescu et al., 2020b) . This procedure produced ∼200k pseudo-annotations on image fields from the Ibadan(Manescu et al., 2020b) and Chittagong-1(Kassim et al., 2021) datasets.

**Hard negative mining.** Negative samples often contain parasite-like artifacts such as platelets and stain precipitates. We performed hard-negative mining by applying the MIL classifier to fields from negative bags and computing CAMs with the positive-class weights. Although these fields do not contain any parasites, the resulting CAMs identify texture patterns the model mistakenly finds discriminative (Sup. Fig. 1). Peaks extracted from these CAMs are converted into bounding boxes using the same procedure as ICAM. This procedure yielded ∼60k hard negative annotations. Although illustrated here for malaria microscopy, the hard negative mining strategy addresses visually confusing but non-pathological patterns, and the underlying idea may extend to other medical imaging tasks, with task-specific adaptations.

### 2.4. Parasite detector training and pseudo-label refinement

The set of ICAM-generated parasite and artefact pseudo-annotations were next used to train an initial one-stage RetinaNet with a ResNet-50 backbone object detector (Lin et al., 2017) $\mathcal{D}_{\text{MILCA-raw}}$. We further incorporated a bootstrapped refinement stage. $\mathcal{D}_{\text{MILCA-raw}}$ is applied to all training fields, and its predictions are retained if their confidence exceeds 0.7. A second detector, $\mathcal{D}_{\text{MILCA}}$, was next trained on the refined pseudo-labels.

The MILCA detectors were trained with a focal loss (Lin et al., 2017) for 30 epochs using an Adam Optimizer with learning rate of $5 \times 10^{-4}$.

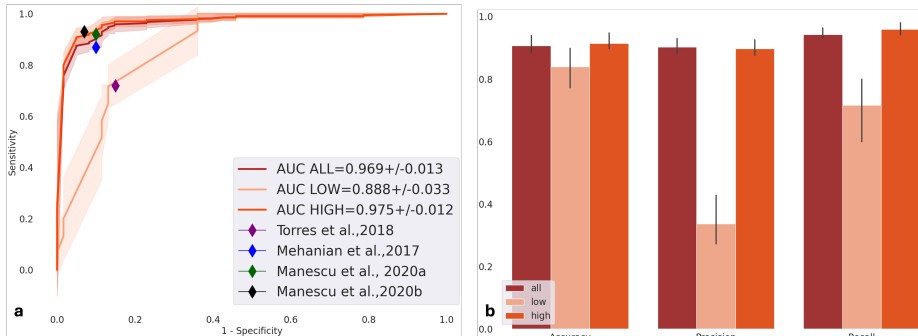

Figure 3: Sample-level classification results. (a) ROC curve with mean AUC over 3 folds ($n = 99$). LOW: <1000 parasites/$\mu$l. Comparison with previous studies is just indicative as the train or test datasets were not made available.(b) Additional sample-level metrics.

## 2.5. Fine-Tuning with Limited Manual Annotations

To evaluate annotation efficiency, we further fine-tuned $\mathcal{D}_{\text{MILCA}}$ using varying fractions of the available manually annotated data (5-75%). For each fraction, we randomly sampled the corresponding subset of annotations three times and report the mean performance across the three runs to reduce sensitivity to the particular subset chosen. Fine-tuning updates only the classification and regression heads and is performed for 10 epochs.

This hybrid regime differs from conventional semi-supervised pipelines, which require manual labels to bootstrap the initial detector. In contrast, MILCA first constructs a detector entirely from weak supervision and incorporates expert annotations only at the final refinement stage.

We compared MILCA fine-tuning with a semi-supervised learning (SemiSL) baseline based on naive pseudo-labeling previously used in microscopy (Shokrollahi et al., 2024). In this setup, a vanilla object detector was first trained on the same fractions of expert annotations as MILCA. This detector was then applied to the remaining unlabeled training images, and only predictions with a high confidence score (above 0.7) were retained as pseudo-annotations. A new detector was subsequently trained from scratch using both the original manual annotations and the generated pseudo-labels.

## 3. Results

### 3.1. Sample-level predictions

The MILCA classifier accurately distinguished malaria-positive from negative samples across all three datasets. In 3-fold cross-validation, it achieved an average AUC of $0.969 \pm 0.013$ (Fig. 3a), with overall accuracy exceeding 90%. Performance was lower on samples with medium to low-parasitemia (<1000 parasites/$\mu$l), reflecting the inherent difficulty of detecting sparse parasite instances (Fig. 3b). These sample-level predictions are produced

without any manual parasite annotations and provide the image field-selection signal used by ICAM.

## 3.2. Weakly supervised parasite detection

Parasite detection performance is evaluated using Average Precision (AP), defined as the area under the precision–recall curve computed over parasite detections at a fixed Intersection-over-Union (IoU) threshold. IoU is defined as the ratio between the area of overlap and the area of union of a predicted bounding box and its corresponding ground-truth box (Everingham et al., 2010). Across the Ibadan dataset (23 test fields, $\sim$700 parasites), Chittagong-1 dataset (10,598 parasites in test set), and secondary detection-only Chittagong-2 dataset (84,961 parasites) a detector trained solely on ICAM pseudo-labels (no manual boxes) achieved mean AP values between 0.1–0.3 at an $IoU = 0.5$ demonstrating that sample-level supervision alone provides sufficient signal to bootstrap a malaria parasite detector. As expected given the small parasite size, AP was higher at lower IoU thresholds (Fig. 5c). Qualitative examples illustrate that MILCA detects many parasites without supervision, and fine-tuning improves recall and bounding-box accuracy (Fig. 5a-b).

## 3.3. Label efficiency and hybrid fine-tuning

We next evaluated parasite detection using an object detector trained on MILCA pseudo-annotations and fine-tuned with different fractions of manually labeled data (5-75%). We compared against a fully supervised detector trained on all ground-truth annotations (vanilla).

Fine-tuning with small fractions of manual annotations yielded substantial gains (Fig. 4 and Fig. 5). Across Ibadan, Chittagong-1, and the secondary *detection-only* Chittagong-2 dataset, MILCA outperformed both the fully supervised detector and a naive pseudo-labeling SemiSL baseline at low budgets (For more details, see Sup. Fig. 3). With only 5% of the labels, MILCA improved AP by 10–20 points over supervised training, reflecting the stronger initialization produced by ICAM pseudo-labels. At 50% labels, MILCA remained competitive or superior across all dataset (Table 2). Evaluated on Chittagong-2—unseen during training—MILCA generalized better than the supervised or naive SemiSL detectors. This suggests that training on large numbers of weak, diverse pseudo-labels provides robustness to staining variability and acquisition differences, which are common in routine malaria microscopy. ICAM-derived pseudo-labels are coarse, as they are generated solely from sample-level supervision, and are not expected to yield precise bounding boxes in isolation. Nevertheless, they provide a dense and task-aligned initialization that enables the detector to reliably identify parasite regions. Consequently, performance under fully weak supervision is more limited by localization accuracy rather than by parasite discovery, and even a small amount of expert fine-tuning substantially improves bounding-box regression, leading to large gains at stricter IoU thresholds.

## 3.4. Parasite count estimation

MILCA produced accurate parasite counts on internal test sets, achieving $R^2 = 0.80$ on Chittagong-1 (Fig. 6). Performance decreased on the secondary *detection-only* Chittagong-2 dataset ($R^2 = 0.14$), consistent with domain shift effects, yet MILCA remained more accurate than the vanilla parasite detector under all annotation budgets. All models tended to

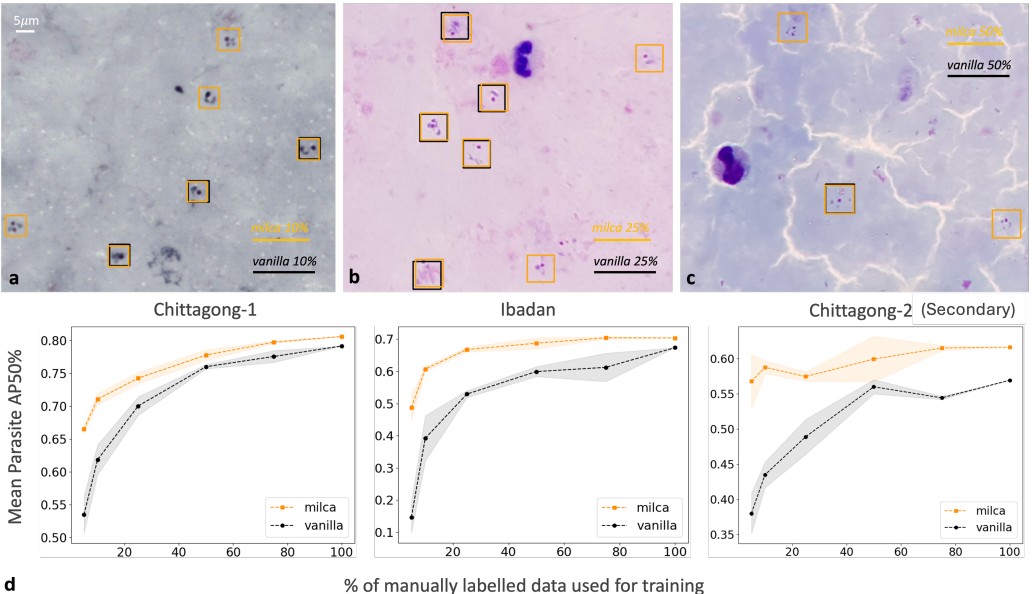

Figure 4: Example detections using the Vanilla and the MILCA and fine-tuned parasite detectors on (a) Ibadan, (b) Chittagong-1 and (c) Chittagong-2 test images. (d) Parasite detection performance. Mean AP at IoU=0.5 for different fractions of labeled data on Chittagong-1, Ibadan, and Chittagong-2 datasets. Experiments repeated 3 times with random fractions.

Table 2: Detection performance (Parasite AP50%, mean with std in parentheses) for Vanilla Supervised, SemiSL (Shokrollahi et al., 2024), and MILCA across three datasets under 5% and 50% annotation budgets. Best performance for each dataset and budget is highlighted in bold.

| Annotation Budget | Method | Ibadan (Internal) | Chittagong-1 (Internal) | Chittagong-2 (Secondary det-only) |
|---|---|---|---|---|
| 5% | Vanilla Supervised | 0.15 (0.05) | 0.53 (0.03) | 0.38 (0.03) |
|  | SemiSL | 0.31 (0.05) | 0.56 (0.03) | 0.33 (0.02) |
|  | MILCA | **0.49 (0.04)** | **0.66 (0.01)** | **0.57 (0.04)** |
| 50% | Vanilla Supervised | 0.60 (0.02) | 0.76 (0.00) | 0.56 (0.01) |
|  | SemiSL | 0.64 (0.01) | 0.75 (0.01) | 0.51 (0.01) |
|  | MILCA | **0.69 (0.00)** | **0.78 (0.01)** | **0.60 (0.03)** |

underestimate counts at very high parasitemia levels, where overlapping parasites introduce ambiguity. Overall, MILCA demonstrates that meaningful object-level supervision can be derived entirely from sample-level diagnostic labels, enabling strong detection and counting performance with minimal expert annotation.

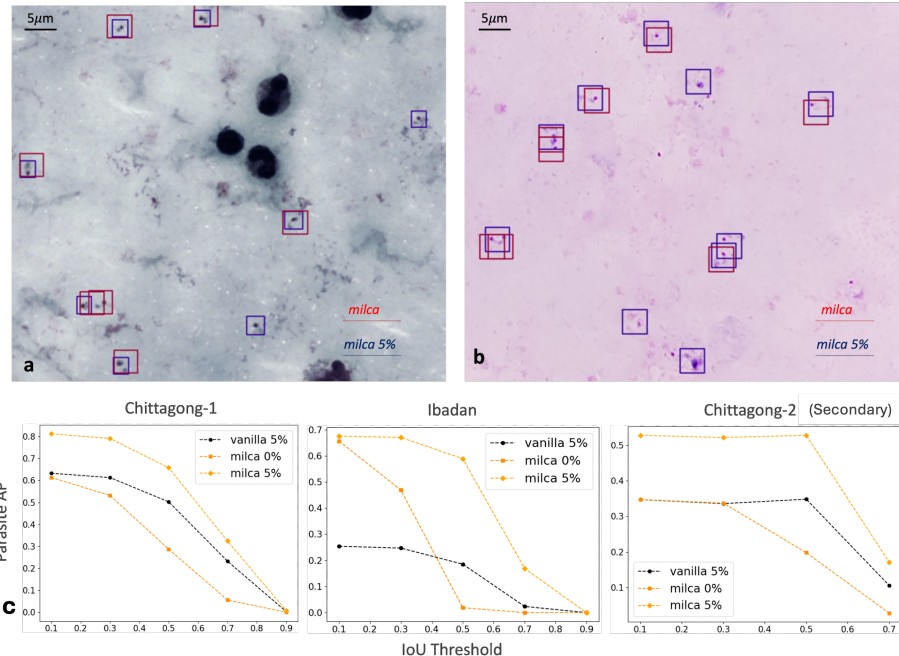

Figure 5: Example detections using the MILCA object detector (red: no fine-tuning; blue: with 5% fine-tuning) on (a) Ibadan and (b) Chittagong-1 test images.(c)Parasite detection performance. AP as a function of IoU threshold.

## 4. Discussion

We introduced MILCA, a weakly supervised framework for malaria parasite detection in thick blood films that relies solely on sample-level diagnostic labels. By combining MIL image field selection, ICAM refinement, and hard-negative mining, MILCA generates dense pseudo-annotations that enable training a practical detector without any bounding-box labels. Across multiple datasets, MILCA achieves competitive detection and counting performance, under fully weak supervision and substantially outperforms supervised and SemiSL baselines when annotation budgets are low. A key finding is that weak supervision *before* strong supervision is highly effective: detectors initialized from ICAM pseudo-labels require far fewer manual annotations to reach high accuracy compared to conventional semi-supervised pipelines. This offers a scalable path for developing diagnostic models in settings where expert time is limited and annotation costs are prohibitive.

From a deployment perspective, MILCA substantially lowers the barrier to developing parasite detection and counting systems by eliminating the need for large-scale parasite-level annotations. The framework relies only on sample-level diagnostic labels (positive/negative), which are already generated as part of routine clinical workflows, and can therefore be applied retrospectively to existing datasets without additional annotation burden.

**Future work.** A limitation of this study is that it does not report strict site-held-out (Ibadan vs Chittagong) evaluation for parasite detection; assessing cross-site generalization

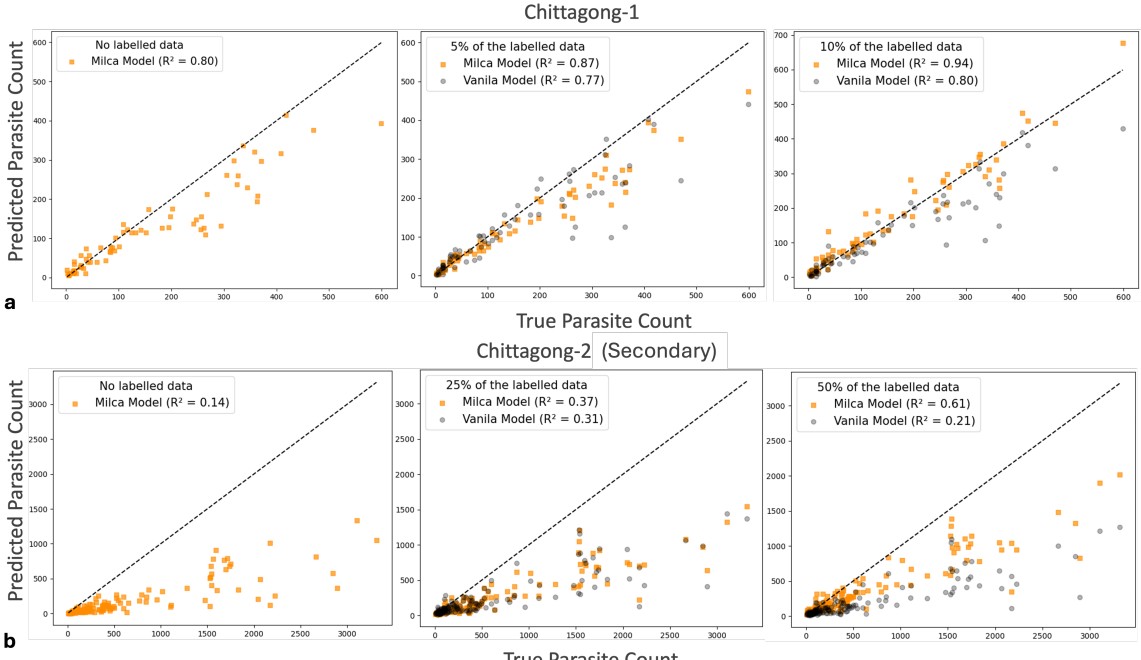

Figure 6: Parasite count evaluation on (a) Chittagong-1 (internal) and (b) Chittaging-2 (secondary detection-only). Scatter plots of predicted vs. true counts for MILCA with and without fine-tuning. MILCA outperforms vanilla detectors under the same annotation budget.

under weak supervision remains an important direction for future work. Future work will also explore more modern tile encoders and MIL aggregation mechanisms, which may further improve sample-level discrimination and the quality of the pseudo-annotations. We will equally investigate integrating MILCA with more advanced semi-supervised detection frameworks, using MILCA-derived pseudo-labels to bootstrap or guide teacher–student training under limited annotation budgets. Enhancements may also include incorporating uncertainty-aware pseudo-label filtering, joint MIL–detector training and contrastive pre-training to mitigate domain shift. MILCA could also be extended to other pathology tasks where annotations are scarce but sample-level labels are readily available.

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

## Appendix A. Probability of Missing Parasites Under Stochastic Tile Sampling

We analyze the probability that a malaria-positive sample yields no parasite-containing tiles under the stochastic bag construction used for Multiple Instance Learning (MIL) training. This analysis is performed on the Chittagong datasets, for which dense parasite bounding-box annotations are available. An equivalent analysis for Ibadan is not possible due to the absence of field-level annotations.

### Bag Construction

Each sample is represented as a bag of 50 image tiles. For a sample containing $F$ image fields ($F = 20$ for Chittagong), one random $224 \times 224$ crop is first extracted from each field. The remaining $m = 50 - F$ tiles are obtained by sampling fields uniformly with replacement and extracting additional random crops from the selected fields.

### Analytical Miss-Rate

Let $r_j$ denote the probability that a single random crop from field $j$ contains at least one parasite, and let $a_j = 1 - r_j$. Under the above sampling scheme, the probability that none of the 50 tiles in a bag contain a parasite is given by:

$$
P_{\mathrm{miss}} = \left( \prod_{j=1}^{F} a_j \right) \left( \frac{1}{F} \sum_{j=1}^{F} a_j \right)^m . \tag{6}
$$

When only the number of parasites $N_j$ in field $j$ is available, $r_j$ can be approximated assuming a uniform spatial distribution of parasites within the field:

$$
r_j \approx 1 - \left( 1 - \frac{A_{\mathrm{crop}}}{A_{\mathrm{field}}} \right)^{N_j} \approx 1 - \exp\left( -N_j \frac{A_{\mathrm{crop}}}{A_{\mathrm{field}}} \right), \tag{7}
$$

where $A_{\mathrm{crop}}$ and $A_{\mathrm{field}}$ denote the areas of a crop and an image field, respectively.

### Monte Carlo Estimation

To obtain an empirical estimate of $r_j$, we perform a Monte Carlo simulation using the available bounding-box annotations. For each field, we randomly sample crop locations uniformly over the valid image area and record the fraction of crops that intersect at least one parasite bounding box. This yields an empirical estimate $\hat{r}_j$ for each field, which is substituted into Eq. (1) to compute $P_{\mathrm{miss}}$ per sample.

### Analysis

Using the available Chittagong annotations, we quantified the probability that a malaria-positive sample yields no parasite-containing tiles under our sampling strategy and found that with a bag size of 50 the median miss-rate is approximately 1–2% (Sup. Table 1), with close agreement between empirical and analytical estimates, supporting the robustness of our tile sampling choice.

Sup. Table 1: Probability of missing all parasite-containing tiles ($P_{\mathrm{miss}}$) for different bag sizes on the Chittagong dataset. Results are summarized across positive samples using both Monte Carlo (MC) estimation based on bounding-box annotations and an analytical approximation based on parasite counts per field. P95 denotes the 95th percentile across samples, i.e., a worst-case estimate for the majority (95%) of samples.

| Bag size | Monte Carlo $P_{\mathrm{miss}}$ | | | Analytical $P_{\mathrm{miss}}$ | | |
|---|---|---|---|---|---|---|
| | Mean | Median | P95 | Mean | Median | P95 |
| 20 | 0.2381 | 0.1822 | 0.6185 | 0.2380 | 0.1729 | 0.6390 |
| 30 | 0.1490 | 0.0783 | 0.4872 | 0.1545 | 0.0724 | 0.5113 |
| 40 | 0.0991 | 0.0336 | 0.3838 | 0.1065 | 0.0303 | 0.4093 |
| 50 | 0.0676 | 0.0145 | 0.2936 | 0.0752 | 0.0127 | 0.3143 |

While an equivalent estimate is not possible for Ibadan due to missing field-level boxes or parasite counts, the Chittagong analysis provides a quantitative reference for the effect of bag size on miss probability.

## Appendix B. Supplementary Figures

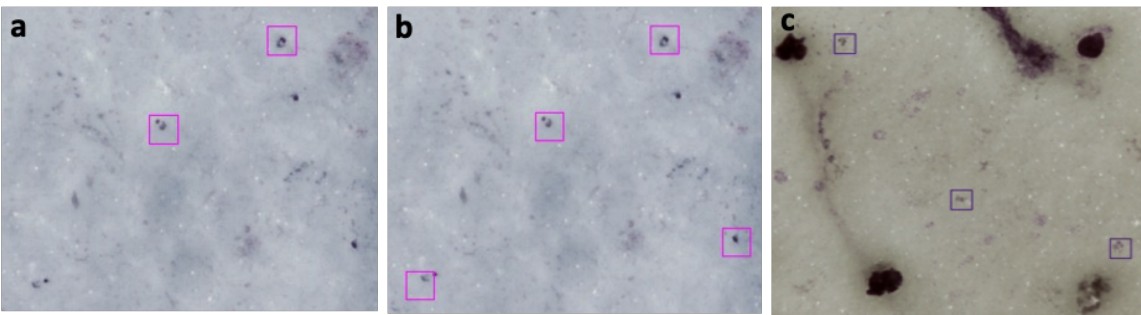

Sup. Fig. 1: (a) Malaria parasite partial annotations generated with CAM. (b) Malaria annotations augmented with ICAM. (c) Hard negative annotations generated by CAM.

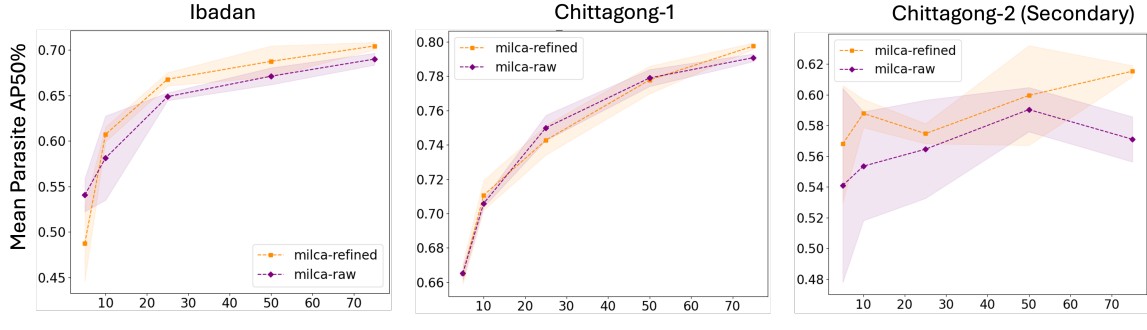

% manually labelled data used for fine-tunning

Sup. Fig. 2: Parasite detection performance. Mean AP at IoU=0.5 for different fractions of labeled data on Chittagong-1, Ibadan, andChittagong-2 datasets. Experiments repeated 3 times with random fractions. Milca-refined refers to the bootstrapped pseudo-label retrained model.

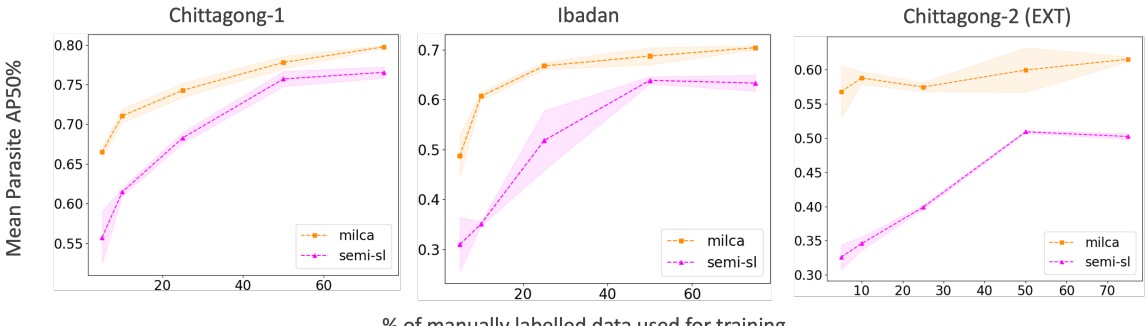

Sup. Fig. 3: Comparison with a naive pseudo-labeling SemiSL baseline. MILCA fine-tuning achieves higher AP across all fractions of labeled data.

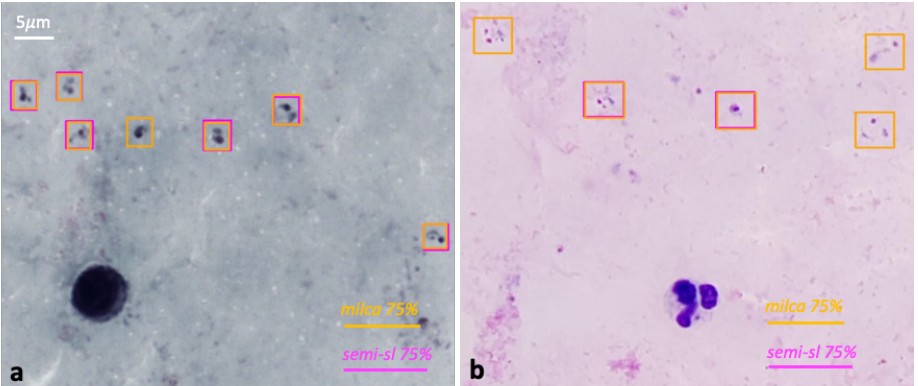

Sup. Fig. 4: Example detections using MILCA fine-tuned and the Semi Supervised object detectors on (a) Ibadan and (b) Chittagong-1 test images.

## Appendix C. MIL Aggregation Ablation

We compare max pooling with an attention-based MIL aggregation mechanism (Ilse et al., 2018) for sample-level malaria classification. Both models use the same encoder, training protocol, and data splits. Results are reported as mean ± standard deviation across runs.

Sup. Table 2: Comparison of MIL aggregation strategies.

| Aggregation | Accuracy | Precision | Recall | AUROC |
|---|---|---|---|---|
| Max pooling | $0.91 \pm 0.02$ | $0.90 \pm 0.02$ | $\mathbf{0.94 \pm 0.02}$ | $0.97 \pm 0.01$ |
| Attention pooling | $0.89 \pm 0.01$ | $\mathbf{0.94 \pm 0.04}$ | $0.84 \pm 0.02$ | $0.95 \pm 0.01$ |

Overall performance is comparable across aggregation strategies. Max pooling yields higher recall, consistent with the presence of sparse positive instances, while attention pooling favors precision. Given the focus of this work on label-efficient weak-to-strong supervision rather than MIL architecture optimization, we used max pooling in all subsequent experiments.

