# OpenReview forum: "MILCA: Malaria Parasite Detection from Sample-Level Weak Labels"
_MIDL.io/2026/Conference — MIDL 2026 Poster_

### Official Review · Reviewer_z1Qy · 2026-01-03

**Confidence:** 3
**Preliminary Rating:** 4
**Final Rating:** 4

**Summary:**

The authors proposed a framework combining Multiple Instance Learning (MIL) for sample-level classification and an iterative Class Activation (CA) Mapping for bounding box learning based on activation maps. The experiments include multiple datasets and provide convincing performance improvement compared with the baseline results.

**Strengths:**

+ The topic of this manuscript is timely and interesting.
+ The proposed method is sound. The design of the iterative CAM based on the activation map is interesting.
+ The datasets involved in the experiments are multiple datasets.

**Weaknesses:**

Major:
- VGG-16 is an outdated model. Please explain why VGG-16 was used? In the Results section, comparative evaluations against other CNN-based models and even transformer-based models are expected.
- The proposed framework includes specialized processing dedicated to the Malaria Parasite images (mainly in Section 2.3, Hard Negative Mining) to enhance the final detection performance. However, it remains unclear whether this specific strategy can be effectively generalized or extended to other types of medical imaging tasks.
- In the abstract, the authors state that traditional manual annotation is not deployment-friendly. Please further discuss how the proposed method improves deployment feasibility in real-world clinical settings.

Minor:
- A reference is missing on Page 2: "fine-tuning (Fig. ??)".

**Detailed Comments:**

The authors can consider using some table-based results. Graph-based or chart-based results make it hard to identify the subtle improvement.

**Justification Of Final Rating:**

Thanks for the authors' response. The authors have covered all my concerns, including major concerns and minor concerns. I keep my original rating due to some obvious limitations in the current manuscript. But now it is overall a good paper.

**Justification Of The Preliminary Rating:**

The method of this manuscript is sound, showing the authors' strong experience with Malaria Parasite images. The results are convincing, even though there are some concerns. This manuscript is easy to follow.

**Questions To Address In The Rebuttal:**

Please address all concerns in "Weaknesses".

---

> ### Author Response · Authors · 2026-01-24
> **Response to Reviewer z1Qy**
>
> We thank the reviewer for summarizing our work and providing valuable feedback We address the weaknesses, detailed comments, and rebuttal questions below:
>
> **Use of VGG-16 and Lack of Comparisons to Modern Backbones**
>
> We acknowledge that VGG is not a state-of-the-art architecture. Our choice was motivated by practical considerations rather than architectural novelty. Our goal in this work is not to benchmark CNN or transformer backbones, but to demonstrate a label-efficient weak-to-strong supervision pipeline that converts sample-level diagnostic labels into object-level supervision. The MILCA framework is agnostic to the choice of encoder, and more modern CNN or transformer-based backbones could be substituted without modifying the pipeline. We now explicitly acknowledge this limitation and identify the evaluation of more recent architectures as future work in the revised manuscript.
>
> **Generalizability of Hard Negative Mining Beyond Malaria**
>
> We thank the reviewer for raising this important question. While the hard negative mining strategy is motivated by malaria microscopy, where common confounders include platelets, stain precipitates, and debris, the underlying principle is not malaria-specific. Specifically, the approach leverages samples known to be negative at the sample level to identify visually salient false positives and explicitly incorporate them as negatives during detector training.
>
> In many medical imaging tasks, including histopathology and cytology, there exist visually salient structures or regions that closely resemble pathological findings but are benign (e.g., reactive tissue, atypical but non-malignant cells). Such look-alike patterns frequently give rise to systematic false positives in object detection models. The proposed hard negative mining mechanism leverages weak supervision from negative samples to explicitly suppress these recurring false positives. We emphasize that the concrete definition of “hard negatives” is task- and modality-specific, and adapting this strategy to other applications would require domain knowledge, but the underlying concept is not limited to malaria. We have added the follwing clariifcation in Section 2.3: **Although illustrated using malaria microscopy, the hard negative mining strategy addresses visually confusing but non-pathological patterns, and the underlying idea may extend to other medical imaging tasks, with task-specific adaptations.**
>
> **Deployment Feasibility and Clinical Relevance**
>
> We appreciate the reviewer’s request for clarification on deployment feasibility. The proposed approach improves deployment readiness by substantially reducing annotation requirements, relying only on sample-level diagnostic labels that are already produced in routine clinical workflows. This eliminates the need for large-scale expert bounding-box annotation, which is often infeasible in resource-limited settings.
> In practice, MILCA enables the development of parasite detection and counting models with minimal additional annotation effort, making it more compatible with real-world clinical deployment, particularly in endemic regions. We have expanded the discussion (section 4) in the revised manuscript to better articulate this practical advantage: **From a deployment perspective, MILCA substantially lowers the barrier to developing parasite detection and counting systems by eliminating the need for large-scale parasite-level annotations. The framework relies only on sample-level diagnostic labels (positive/negative),
> which are already generated as part of routine clinical workflows, and can therefore be applied retrospectively to existing datasets without additional annotation burden.**
>
> **Presentation and Minor Issues**
>
> We have fixed the missing figure reference (“fine-tuning (Fig. ??)”) and addressed all minor presentation issues.

---

### Official Review · Reviewer_RMzm · 2026-01-10

**Confidence:** 4
**Preliminary Rating:** 3
**Final Rating:** 3

**Summary:**

In MILCA, the authors address the annotation bottleneck in malaria parasite detection by learning from sample-level diagnostic labels (positive/negative) instead of costly bounding box annotations. The method combines MIL (Multiple Instance Learning) with an iterative Class Activation Mapping (ICAM) procedure, which progressively identifies parasite locations through erasure and refinement, that generates pseudo-labels to train an object detector.

**Strengths:**

1. The motivation is clear in the paper. The authors address a genuine clinical bottleneck in this work, since existing malaria detection systems require tens of thousands of manual bounding box annotations, being very impractical in resource-limited settings in the countries where malaria is most prevalent.

2. Standard CAMs highlight only the most prominent instance, but malaria parasites are small, numerous, and faint. The ICAM progressively reveals multiple parasites, and hard negative mining from parasite-free samples addresses artifacts, such as platelets and stain precipitates.

3. With only 5% of annotations, authors show that MILCA improves AP by 10-20 points over supervised baselines trained on the same budget

**Weaknesses:**

1. The jump from "weak supervision alone" to "5% fine-tuning" shows 2-3× improvement, suggesting the pseudo-labels are quite noisy? Respected authors, please comment on it.

2. Some incomplete baselines:
a) No comparison to other WSOD methods adapted for microscopy
b) Any reason why the Semi-supervised baseline (SemiSL) is very simple (naive pseudo-labeling)? Why not use any modern SSL methods (e.g. ref1: Teacher-student pseudo-labeling, ref2: Contrastive pre-training)

3. The Ibadan detection test set is tiny (23 fields, ~700 parasites). With so few test fields, even small changes in thresholding or annotation noise can significantly impact AP (maybe confidence interval should be reported).

References:
1. Ref1: MIAdapt: Source-free Few-shot Domain Adaptive Object Detection for Microscopic Images (Dilawar et. al. 2025).
2. Ref2: Domain-Aware Self-Supervised Learning for Cell-Level Malaria Classification (Tchinda et. al. 2025).

**Detailed Comments:**

1. Figure references missing in multiple places(e.g. 1st paragraph of Methods section, 2nd paragraph section 3.3) Please fix them. It becomes very tough to connect the figure with the corresponding text without incomplete reference.

2. There might be a typo of "typically" in the introduction section. Please fix it.

3. Based on the paper context, "AP" seems to stand for Average Precision. But it's not defined anywhere in the paper. The same applies to "IoU"; there should be a clarification or a reference. Defining key metrics is important, especially for new readers in the field.

**Justification Of Final Rating:**

Thanks for adding the clarification to the uncertainty of weak supervision versus fine-tuning, and on the latest baselines. They also fixed their presentation issues to some extent. However, I would really want the author(s) to work on the semi-supervised detection as a baseline or a modification. Also, for Ibadan Detection Test Set, reporting "mean performance" is misleading since it's already a small dataset and a small outlier can shift the distribution heavily.

So, I am sticking to my "Borderline" final rating decision for now.

**Justification Of The Preliminary Rating:**

The paper addresses a clinically important problem and demonstrates strong label efficiency gains (10-20 AP improvement with only 5% annotations), with ICAM being a sensible adaptation for revealing multiple small parasites.

However, the manuscript appears incomplete, with missing figure references (e.g., pages 3, 7), undefined key metrics (e.g. AP, IoU), and typos that require correction before publication. More critically, the experimental evaluation has some gaps: the Ibadan test set is very small (23 fields, ~700 parasites) without confidence intervals, the semi-supervised baseline is overly simplistic without justification for not using more up-to-date methods, and there's no comparison to other WSOD methods adapted for microscopy.

While the core ideas have merit and the multi-dataset evaluation shows promise, the presentation quality and experimental rigor need strengthening. I'm willing to raise the score if the authors address these issues and provide clearer experimental justification.

**Questions To Address In The Rebuttal:**

Please answer my queries in the Strengths and Detailed Comments section above

---

> ### Author Response · Authors · 2026-01-24
> **Response to Reviewer RMzm**
>
> We thank the reviewer for summarizing our work and providing valuable feedback We address the weaknesses, detailed comments, and rebuttal questions below:
>
> **Weak supervision versus fine-tuning**
>
> We thank the reviewer for highlighting this uncertainity.
>  The ICAM-generated pseudo-labels are intentionally coarse and noisy, as they are derived entirely from sample-level supervision without any manual bounding boxes. Their primary role is to provide a dense, task-aligned initialization for the detector rather than precise localization.
> In practice, detectors trained solely on pseudo-labels already learn to identify parasite locations reasonably well (Fig. 5), but the resulting bounding boxes are often spatially imprecise, reflecting the coarse nature of CAM-based supervision which disproportionately affects AP at stricter IoU thresholds, even when object presence is correctly detected. The followig clarification has been added to setcion 3.3:**ICAM-derived pseudo-labels are coarse, as they are generated solely from sample-level
> supervision, and are not expected to yield precise bounding boxes in isolation. Nevertheless,
> they provide a dense and task-aligned initialization that enables the detector to reliably
> identify parasite regions. Consequently, performance under fully weak supervision is more
> limited by localization accuracy rather than by parasite discovery, and even a small amount
> of expert fine-tuning substantially improves bounding-box regression, leading to large gains
> at stricter IoU thresholds.**
>
> **Baselines: WSOD, Semi-Supervised, and Related Methods**
>
> We thank the reviewer for raising this important point and for highlighting relevant recent work. Many WSOD methods assume image-level labels where each labelled image contains at least one object instance, often coupled with proposal-based pipelines or region-level pretraining. This assumption does not hold in thick blood film microscopy, where a malaria-positive sample may consist of tens to hundreds of fields and only a small fraction contains parasites.  Instead, MILCA is specifically designed for the malara detection clinical workflow by combining sample-level MIL with CAM-based pseudo-label generation.
>
> **Semi-supervised detection.**
>
> We acknowledge that more advanced semi-supervised detection methods (e.g., teacher–student or consistency-based frameworks) could provide stronger baselines. In this work, we intentionally adopted a simple and widely used naive pseudo-labelling strategy as a reference, as our primary objective is not to benchmark semi-supervised methods, but to evaluate label efficiency when only sample-level diagnostic labels are available to initialize detection.
> We agree, however, that integrating MILCA with more advanced semi-supervised frameworks, using MILCA-derived pseudo-labels as a strong initialization or teacher signal is a promising direction. We now explicitly acknowledge this in the Future Work section of the revised manuscript.
>
> **Self-supervised learning and domain adaptation.**
>
> We also now explicitly acknowledge and discuss recent microscopy-focused self-supervised and domain adaptation methods (MIAdapt; Dilawar et al., 2025. – we could not find Ref2) in the revised manuscript, in the introduction: **Related efforts have explored self-supervised representation learning and domain adaptation for  microscopy analysis, aiming to improve robustness across imaging conditions or reduce labeled data requirements (Dilawar et al. 2025)** .
>
> **Size of the Ibadan Detection Test Set**
>
> We agree that the Ibadan detection test set is small (23 fields, approximately 700 parasites), which can increase sensitivity to annotation noise and evaluation thresholds. While we do not report formal confidence intervals in the current submission, we mitigate variance by repeating label-fraction experiments with three random subsets and reporting mean performance. Moreover, our key conclusions are corroborated on the substantially larger Chittagong-1 test set and on the fully held-out Chittagong-2 detection-only evaluation. We will revise the manuscript to explicitly note the limited size of the Ibadan test set and to emphasize conclusions supported by the larger evaluations.
>
> **Key metric definition**
>
> We thank the reviewer for pointing this out. We agree that key evaluation metrics should be explicitly defined for clarity.
> We have added the following clarification in section3.2: **Parasite detection performance is evaluated using Average Precision (AP), defined as the area under the precision–recall curve computed over parasite detections at a fixed Intersection-over-Union (IoU) threshold. IoU is defined as the ratio between the area of overlap and the area of union of a predicted bounding box and its corresponding ground-truth box.(Evernigham et al., 2010)**
>
> **Presentation and minor issues**
>
> We have fixed the missing figure references together with other minor issues.

---

### Official Review · Reviewer_RVKf · 2026-01-13

**Confidence:** 4
**Preliminary Rating:** 4
**Final Rating:** 4

**Summary:**

MILCA converts sample-level malaria labels into object-level parasite detections in thick blood films via a three-stage pipeline: (i) a MIL classifier with a ConvNet encoder (VGG-16) and max feature pooling to predict sample positivity and score fields, (ii) ICAM, an iterative CAM refinement that aggregates CAMs with decay, and (iii) training a RetinaNet detector (with ResNet-50 backbone) on parasite + hard-negative pseudo-boxes, optionally fine-tuned with small fractions of expert boxes. The experiments show strong label efficiency, by initializing from ICAM pseudo-labels and fine-tuning with 5% labels improves detection over fully supervised and naive SemiSL baselines, and the same holds at 50%. The work is significant as a practical weak-supervision route to detection and parasite counting where dense manual annotations are expensive and laborious.

**Strengths:**

- The proposed practical supervised setting relies on sample-level diagnostic labels which aligns with malaria workflows.
- The idea of MIL-based field selection for "filter first" + CAM pseudo annotation is very interesting and might help in other medical workflows with high-resolution images as well.
- The paper shows solid label-efficiency evidence, even at low budgets (e.g. 5%), MILCA improves substantially over vanilla supervsied and the provided SemiSL baseline across datasets.
- The work includes a multi-dataset evaluation (of the same task), with different settings and a external testset. This is a strong validation setup of the proposed method. However, please also see weakness on the exact data splits used.

I think the community can benefit from this idea also for other setups using high-resolution medical images (even if not validated in the paper).

**Weaknesses:**

- Ambiguity in external validation: Chittagong-2 is used exclusively for external evaluation, but it is positive only and likely closely related in acquisition domain of Chittagong-1. This weakens the generalization claims without stricter site-held-out experiments.
- The MIL aggregation choice is not justified. The MIL classifier uses max feature pooling because positives may be sparse but no comparison is shown against other pooling strategies (e.g. Attention-based MIL, Ilse et al 2018), that often improves MIL performance and can provide instance (which tiles from the image) relevance.
- The sampling description is potentially inconsistent, training uses bags of 50 randomly cropped tiles. However, its unclear what the miss-rate is for parasite counting fields under this sampling.
- Baselines are limited and comparisons contain mainly a vanilla supervised detector and a naive pseudo-label SemiSL baseline. Next, stronger MIL baselines could clarify whether gains come from CAM + hard negatives vs simpler alternatives.
- Some presentation issues, e.g. "Fig. ??" placeholders and minor typos appear throughout the paper. Another editing round with peers could solve this.

**Detailed Comments:**

- Please clarify dataset provenance and what “external” means operationally. Also add a stricter split such as train Ibadan -> test Chittagong-1/2 (and the reverse) to actually show generalization
- Clarify bag construction precisely per dataset: number of fields used, how 50 tiles are sampled when only 20 fields exist, and whether sampling uses replacement or multiple crops per field.
- Add a small ablation: max pooling vs attention pooling (or another learned aggregator) and report impact on (i) sample AUC and (ii) downstream tasks.
- Minor: fix figure numbering and typos

**Justification Of Final Rating:**

The authors have addressed some of my questions. Due to the lack of comparison methods, the old feature encoder without any permutations of methods and the missing real external evaluation, I keep my score.

**Justification Of The Preliminary Rating:**

The overall approach (MIL-based filtering, then detection based on informative tiles) is simple and should be reproducible also for other medical imaging problems involving high-resolution images. However, the paper can benefit from more clarification on method selection (MIL, VGG-16 encoder), data-splitting, more baselines and another dataset with another task to show generalization of the approach beyond Malaria Parasite Detection.

**Questions To Address In The Rebuttal:**

- What is the exact relationship between Chittagong-1 and Chittagong-2 (site, protocol, hardware), and why not report a stricter cross-site split (Ibadan-only training vs Chittagong testing, or vice versa)?
- Can you justify max feature pooling with an ablation against a learned MIL aggregator (e.g., attention pooling), and comment on tile-selection?
- How exactly are the 50 tiles per bag constructed for datasets with 20 fields/sample, and what is the estimated probability that a positive sample yields no parasite-containing tiles under your sampling?
- You use VGG-16 for MIL but ResNet-50 for the detector. Is VGG-16 pretrained, frozen or trained end-to-end, and did you test a more modern encoder for MIL?

---

> ### Author Response · Authors · 2026-01-24
> **Response to Reviewer RVKf**
>
> We thank the reviewer for summarizing our work and for the constructive feedback. We address the main concerns below.
>
> **Relationship between Chittagong-1 and Chittagong-2.**
>
> Chittagong-1 and Chittagong-2 were collected in the same geographic region using similar smartphone-based microscopy setups. The datasets were acquired for different studies at different points in time and exhibit differences in image appearance, albeit modest in comparison to the differences observed between the Ibadan and Chittagong datasets; this has been clarified in the revised manuscript. In addition, Chittagong-2 contains only malaria-positive samples with higher parasite densities than Chittagong-1.
> Chittagong-1 contains both malaria-positive and malaria-negative samples and is used for MIL training and internal evaluation. Chittagong-2 is used exclusively for object-level parasite detection and counting and is never used during MIL training, ICAM pseudo-label generation, or detector initialization. We originally used the term external to indicate that Chittagong-2 is fully held out from all previous stages, rather than to claim strict site-held-out generalization, and agree that this distinction was not sufficiently clear. This has been clarified in **Section 2.1**.
>
> **Scope of generalization claims.**
> We agree that Chittagong-2 does not constitute an independent multi-site validation with both positive and negative samples and therefore does not support strong claims of cross-site generalization. Its role is instead to evaluate object-level detection and counting on a dataset with a different parasite density distribution, without any exposure during training. More broadly, the primary objective of this study is label efficiency rather than domain generalization: to demonstrate that sample-level diagnostic labels can be converted into effective object-level supervision and that detectors initialized from weak supervision require substantially fewer expert annotations than fully supervised or naive semi-supervised alternatives. We revised the manuscript accordingly to avoid claims of strict cross-site or multi-centre validation and to explicitly describe Chittagong-2 as a *secondary detection-only evaluation dataset*.
>
> We agree that strict site-held-out evaluation for both MIL training and parasite detection would provide a stronger assessment of cross-site generalization. However, such experiments would require substantial retraining under alternative splits and were not feasible within the rebuttal timeframe. We explicitly acknowledge this as a limitation in the revised Discussion and identify it as an important direction for future work.
>
> **Tile selection.**
> The following clarification has been added to Section 2.2: **For each sample, we construct a bag of 50 tiles. When
> a sample contains F image fields (F = 20 for Chittagong-1; F = 100 for Ibadan), we first
> extract one random tile from each available field. If F < 50, the remaining (50 − F ) tiles
> are obtained by sampling fields uniformly with replacement, allowing the same field to be
> selected multiple times and extracting additional random crops from the selected fields**
>
> **Tile sampling miss-rate.**
>
> Using the available annotations, we quantified the probability that a malaria-positive sample yields no parasite-containing tiles under our stochastic bag construction. With a bag size of 50, the median miss-rate is approximately 1–2% (Monte Carlo), with a 95th-percentile miss-rate below 30%, and close agreement between empirical and analytical estimates (**Appendix A**).
>
> **MIL aggregation strategy.**
>
> In response to the reviewer’s suggestion, we performed an additional ablation comparing max pooling with attention-based MIL aggregation (Ilse et al., ICML 2018), using the same encoder, training protocol, and data splits. Performance is comparable overall, with max pooling yielding higher recall and attention pooling favoring precision, while AUROC remains similar. In highly sparse settings, attention-based pooling can dilute rare but informative signals, whereas max pooling emphasizes the most discriminative field. Given our focus on label-efficient weak-to-strong supervision rather than MIL architecture optimization, we retain max pooling in the main experiments and report full results in the **Appendix C**.
>
> **MIL encoder choice**
>
> The VGG encoder used for MIL is ImageNet-pretrained and trained end-to-end with no frozen layers; this has been clarified in the manuscript. We did not evaluate more modern MIL encoders due to time and computational constraints. Our goal was not to benchmark MIL architectures, but to establish a practical weak-to-strong supervision pipeline for label-efficient parasite detection. The framework is encoder-agnostic, and incorporating more recent backbones is identified as future work.

---

### Author Rebuttal · Authors · 2026-01-24

**Rebuttal:**

We thank the reviewers for their constructive feedback and positive assessment of the clinical relevance and label-efficiency of MILCA. In response, we have revised the manuscript to improve clarity, scope, and methodological transparency.


Specifically, we:

 (i) clarified the role of each dataset and revised the terminology around “external” evaluation, explicitly positioning Chittagong-2 as a secondary detection-only evaluation dataset and avoiding claims of strict cross-site generalization;

(ii) provided a precise description of the tile sampling strategy and added an analytical and Monte Carlo analysis quantifying the probability of missing parasite-containing tiles, summarized in a new appendix table;

(iii) clarified the behaviour of weak supervision versus limited fine-tuning, explaining that ICAM pseudo-labels provide strong object-level priors but coarse localization, with large gains at 5% labels driven primarily by improved bounding-box regression;

(iv) justified architectural and baseline choices while explicitly identifying modern MIL encoders, semi-supervised methods, and self-/domain-supervised approaches as important future work.

We additionally performed a MIL aggregation ablation comparing max pooling and attention-based pooling, showing comparable performance.

We also addressed concerns regarding the small Ibadan test set by explicitly noting its limited size, emphasizing repeated runs and corroboration on larger Chittagong evaluations, and avoiding over-interpretation of absolute AP values. Finally, we corrected presentation issues (missing figure references, undefined metrics, typos), added table-based summaries, and expanded the Discussion and Abstract to more clearly articulate deployment feasibility in real-world clinical settings.
We believe these revisions strengthen the manuscript while preserving its core contribution: demonstrating a practical, label-efficient pipeline that converts routine sample-level diagnostic labels into effective object-level supervision for malaria parasite detection.

The modifications are highlighted in red in the revised manuscript

**Supporting Material:**

/attachment/e7d30fcacd7809ac952b4f15aec623ba56bdc6d4.pdf

---

### Comment · Area_Chair_gEH6 · 2026-01-28

Dear Reviewers,

We are now in the discussion phase. If you have not yet done so, please check the authors’ rebuttal and evaluate how well your concerns have been addressed. I encourage you to engage in discussion with the authors and other reviewers where helpful.

Most importantly, please update your Final Rating after considering the rebuttal and discussion.

Your input is important for a fair meta-review and final decision. Thank you for your continued effort.

AC

---

### Meta-Review · Area_Chair_gEH6 · 2026-02-01

**Recommendation:** Accept (Poster)
**Confidence:** 4

**Metareview:**

All three reviewers are positive and agree that this paper is well-motivated and practically valuable for malaria parasite detection under sample-level weak supervision. The main strength is a practical pipeline (MIL + CAM-based pseudo-label refinement + detector training) that effectively turns weak labels into object-level detections and demonstrates clear gains under limited annotation budgets. Reviewers also appreciate the additional ablations and clarifications, including the role of the “external” dataset, tile-sampling miss-rate analysis, and MIL aggregation comparisons, which strengthen the empirical evidence and improve transparency. Remaining limitations include that the external validation is not a strict multi-site generalization test and that stronger modern baselines/backbones could further solidify the conclusions. Nonetheless, the method is solid, the problem is important, and the authors’ responses address the key concerns. Overall, I recommend acceptance.

---

### Decision · Program_Chairs · 2026-02-13

Accept (Poster)